# Identification of Hypothalamic Long Noncoding RNAs Associated with Hypertension and the Behavior/Neurological Phenotype of Hypertensive ISIAH Rats

**DOI:** 10.3390/genes13091598

**Published:** 2022-09-07

**Authors:** Larisa A. Fedoseeva, Nikita I. Ershov, Ivan A. Sidorenko, Arcady L. Markel, Olga E. Redina

**Affiliations:** Institute of Cytology and Genetics, Siberian Branch of Russian Academy of Sciences, 630090 Novosibirsk, Russia

**Keywords:** long noncoding RNA, hypothalamus, hypertension, behavior/neurological phenotype, ISIAH rat strain

## Abstract

Long noncoding RNAs (lncRNAs) play an important role in the control of many physiological and pathophysiological processes, including the development of hypertension and other cardiovascular diseases. Nonetheless, the understanding of the regulatory function of many lncRNAs is still incomplete. This work is a continuation of our earlier study on the sequencing of hypothalamic transcriptomes of hypertensive ISIAH rats and control normotensive WAG rats. It aims to identify lncRNAs that may be involved in the formation of the hypertensive state and the associated behavioral features of ISIAH rats. Interstrain differences in the expression of seven lncRNAs were validated by quantitative PCR. Differential hypothalamic expression of lncRNAs LOC100910237 and RGD1562890 between hypertensive and normotensive rats was shown for the first time. Expression of four lncRNAs (Snhg4, LOC100910237, RGD1562890, and Tnxa-ps1) correlated with transcription levels of many hypothalamic genes differentially expressed between ISIAH and WAG rats (DEGs), including genes associated with the behavior/neurological phenotype and hypertension. After functional annotation of these DEGs, it was concluded that lncRNAs Snhg4, LOC100910237, RGD1562890, and Tnxa-ps1 may be involved in the hypothalamic processes related to immune-system functioning and in the response to various exogenous and endogenous factors, including hormonal stimuli. Based on the functional enrichment analysis of the networks, an association of lncRNAs LOC100910237 and Tnxa-ps1 with retinol metabolism and an association of lncRNAs RGD1562890 and Tnxa-ps1 with type 1 diabetes mellitus are proposed for the first time. Based on a discussion, it is hypothesized that previously functionally uncharacterized lncRNA LOC100910237 is implicated in the regulation of hypothalamic processes associated with dopaminergic synaptic signaling, which may contribute to the formation of the behavioral/neurological phenotype and hypertensive state of ISIAH rats.

## 1. Introduction

Recently, in studies on molecular mechanisms underlying the development of socially important diseases, much attention was given to the role of long noncoding RNAs (lncRNAs). lncRNAs are involved both in the regulation of gene transcription and in post-transcriptional events [1] that affect a wide variety of cellular processes, can significantly alter cell functions, and contribute to the development of many pathological conditions [2,3], including cardiovascular diseases [4,5,6]: cardiac remodeling [7], heart failure [8], myocardial infarction [9], and hypertension [10,11,12,13,14,15]. LncRNAs are often regarded as biomarkers, potential therapeutic targets, or precise indicators of disease prognosis [16,17,18].

Despite considerable interest, the understanding of lncRNAs’ biological functions and interactions is still far from being complete [14]. Advances in next-generation sequencing technologies have made it possible to obtain genome-wide expression profiles and identify lncRNA-related protein-coding genes as well as to determine the biological processes regulated by lncRNAs. There is a notion that lncRNAs may participate in multiple metabolic processes [18].

The hypothalamus is the central brain structure involved in the organization and modulation of a neuroendocrine response to various stressful stimuli [19,20] and in the maintenance of the body’s milieu intérieur in fluctuating environments [21,22]. Hypothalamic dysregulation leads to various metabolic and functional disorders, including an increase in blood pressure. The pathogenesis of most forms of hypertension is associated with a wide variety of functional changes in the hypothalamus [23]. The important function of the hypothalamus in the control of various types of behavior is also recognized, e.g., wakefulness, locomotor activity, and foraging [24,25] as well as the development of anxiety-related disorders [26].

ISIAH rats represent a model of a stress-sensitive form of arterial hypertension and are characterized by high reactivity of the hypothalamic–pituitary–adrenal axis and sympathetic–adrenal system [27]. The ISIAH rat strain has been selected for a sharp blood pressure increase under the conditions of short-term restraint stress caused by placing the rat in a tight wire mesh cage for 30 min. In addition to high blood pressure, ISIAH rats exhibit characteristic structural and functional changes in target organs of hypertension [28]. It has been shown that, in the hypothalamus of ISIAH rats, the concentration of noradrenaline is elevated [29], which is known to be associated with the manifestation of negative emotions such as anxiety and/or fear [30]. Compared to control normotensive rats, ISIAH rats show hyper-reactivity in various behavioral tests (open field test, light/dark test, sound stress test, and fatigue test) that assess the locomotor and exploratory activity of rats as well as the level of anxiety (emotionality) in an unfamiliar environment [31,32].

Previously, using transcriptome sequencing (RNA-Seq), we have identified genes that are differentially expressed (DEGs) in the hypothalamus of hypertensive ISIAH rats versus control normotensive WAG rats, and among them, we identified genes associated with the hypertensive status of ISIAH rats and making the largest contribution to interstrain differences [33]. The aim of this work was to identify lncRNAs expressed in the hypothalamus of hypertensive ISIAH and normotensive WAG rats and coexpressed DEGs associated with hypertension and the behavior/neurological phenotype. A subaim was to determine the biological processes and metabolic pathways in which the coexpressed DEGs participate.

## 2. Materials and Methods

### 2.1. Animals

The work was performed on male hypertensive ISIAH/Icgn rats (abbreviation of “inherited stress-induced arterial hypertension”) and normotensive WAG/GSto-Icgn (Wistar Albino Glaxo) rats. The rats were kept in the Vivarium for Conventional Animals (federal research center Institute of Cytology and Genetics, SB RAS, Novosibirsk, Russia) under standard conditions at 22 ± 2 °C on a 12/12 h light/dark cycle (lights on at 8:00 a.m.) with dry laboratory feed and water available *ad libitum*. All rats analyzed in the study were three months old. Rat hypothalami were collected and stored in RNAlater (Qiagen, Chatsworth, CA) at −70 °C until use. For RNA-Seq analysis, three rats were used in each group. Confirmation of the sequencing results was performed by quantitative PCR (qPCR) on larger groups of ISIAH (*n* = 10) and WAG (*n* = 9) rats. The animal experiments were approved by the Institute’s Animal Care and Use Committee. The study protocol was approved by the Bioethical Council of the federal research center Institute of Cytology and Genetics SB RAS (Novosibirsk, Russia), Protocol No. 69 dated 20 January 2021.

### 2.2. RNA-Seq Analysis

The collected samples were sent to JSC Genoanalytica (Moscow, Russia), where RNA-Seq based on mRNA extraction was performed. The details of the protocol have been described earlier [33]. The obtained data were mapped to the RGSC Rnor_5.0\rn5.074 reference genome with the help of the Tophat2 (v2.0.13) aligner [34]. Annotation of transcripts was based on the NCBI Gene/RefSeq database. Cufflinks/Cuffdiff (v.2.2.1) software applications were then employed to quantify mRNA abundance (evaluated as reads per kilobase per million mapped reads, RPKM) and to reveal differentially expressed transcripts. A gene was defined as being expressed if it matched the Cufflinks criteria on suitability for statistical testing (test status ‘OK’). Genes were considered differentially expressed at a false discovery rate (*q*-value) of <5% [35]. The RNA-Seq data were deposited in the NCBI Short Read Archive database under the accession number PRJNA299102.

### 2.3. qPCR for Validation of the RNA-Seq Results

Hypothalamus samples stored in the RNAlater solution were rinsed in saline, and RNA was isolated using the extractRNA reagent (Evrogen, Moscow, Russia). Residual genomic DNA was removed by treatment with DNase I (Promega, Madison, WI, USA) according to the manufacturer’s protocol. Optical density was measured on an Implen NanoPhotometer (Implen, Munich, Germany). In all samples, the ratio of optical density values at 260/280 nm was 1.8–1.9. Reverse transcription was performed using 2 µg of RNA, random primers (N9) (250 pmol), and reverse transcriptase (BIOSAN, Novosibirsk Oblast, Russia) (100 U) in 50 µL of a buffer (Vector-Best, Novosibirsk Oblast, Russia): 1 h at 37 °C, 30 min at 42 °C, 10 min at 50 °C. The reaction was stopped by heating at 75 °C (5 min).

qPCR was conducted as described previously [36] in a 20 μL reaction mixture containing SYBR Green, forward and reverse primers, 1 U of HotStart Taq polymerase (Vector-Best, Russia), and 1 μL of the cDNA template. The primer sequences and concentrations are shown in Appendix A.

Amplification was conducted using an iCycler iQ Real-Time PCR Detection System (Bio-Rad Laboratories, Woodinville, WA, USA) as follows: denaturation for 1 min at 94 °C, followed by 40 cycles of 15 s at 94 °C, 20 s at an annealing temperature, and 20 s at 72 °C. The fluorescent signal was registered at T_acq_ for 10 s, and the melting curve was generated in the temperature range from 65 to 94 °C.

The analyzed cDNAs were quantified using standard calibration curves built by means of dilutions of a “standard” cDNA (a mixture of aliquots of all the cDNAs). A detailed description of the method is given in [37]. qPCR was performed on lncRNAs RGD1562890, small nucleolar RNA host gene 4 (Snhg4), Brain cytoplasmic RNA 1 (Bc1), uncharacterized LOC100910237 (LOC100910237), growth arrest specific 5 (Gas5), uncharacterized LOC100910973 (LOC100910973), uncharacterized LOC100909675 (LOC100909675), and tenascin XA, pseudogene 1 (Tnxa-ps1), using as a normalization factor the geometric mean of the amplification data on three reference genes: peptidylprolyl isomerase A *(Ppia*)*,* hypoxanthine phosphoribosyltransferase 1 *(Hprt*)*,* and non-POU domain containing, octamer-binding *(Nono*). Statistical analysis was performed in Statistica v.10.0 (Statsoft, Tulsa, OK, USA). According to the Kolmogorov–Smirnov test and Shapiro–Wilk *W* test, the distribution of traits under study did not differ from the theoretically expected normal distribution. Statistical significance of differences was assessed by Student’s *t*-test.

### 2.4. Functional Annotation of DEGs

This procedure was performed using the DAVID (Database for Annotation, Visualization and Integrated Discovery) gene annotation tool [38]. The *Rattus norvegicus* genome served as the background list for over-representation analysis. The Gene Ontology (GO) option in DAVID was selected to identify significantly (*p* < 0.05) enriched biological processes. The Neurological Disease Portal in the Rat Genome Database (RGD) [39] (accessed on 23 March 2022) was utilized to identify genes associated with hypertension and the behavior/neurological phenotype, including grooming, abnormal locomotor behavior (vertical activity and hyperactivity), abnormal cognition and abnormal emotion/affect behavior including an abnormal response to novelty. The Atlas of Combinatorial Transcriptional Regulation in Mouse and Man [40] and the Search Tool for the Retrieval of Interacting Genes/Proteins (STRING) database [22] were used to reveal the DEG encoding transcription factor genes. The STRING database was employed for the construction of functional enrichment networks. STRING networks were sent to the Cytoscape_v3.9.1 using StringApp where the functional annotation for the STRING networks was performed.

### 2.5. Correlation Analysis

To identify lncRNA–mRNA coexpression pairs, the data (RPKM values) were log-transformed, centered, and normalized. Pearson’s correlation analysis was performed between the lncRNAs’ expression and the expression profile of the protein-coding DEGs detected in the same samples. According to the Kolmogorov–Smirnov test, the observed distribution of traits did not differ from the theoretically expected normal distribution. Next, 95%, 99%, and 99.9% levels of significance [df = 4; *p* < 0.05 (0.811); *p* < 0.01 (0.917); *p* < 0.001 (0.974)] based on the two-tailed test were assumed to indicate statistical significance of the identified lncRNA–DEG coexpression. Software packages Statistica 7.0 (StatSoft, Tulsa, OK, USA) and JACOBI4 [41] were used for data analysis and presentation.

## 3. Results

### 3.1. LncRNAs Detected as Expressed by RNA-Seq Analysis

In the present experiment, mRNA isolation technology was used to perform the RNA-Seq procedure; therefore, only lncRNAs with a polyA tail were subjected to the analysis. A comparison of the hypothalamic transcriptomes between hypertensive ISIAH and normotensive WAG male rats at the age of three months revealed 22 lncRNAs (Table 1) that were assumed to be expressed because they matched the Cufflinks criteria on suitability for statistical testing (test status ‘OK’). The highest transcription level was detected for lncRNA Bc1. The ISIAH/WAG difference, log_2_(fold_change), was found to be the largest for lncRNA Snhg4.

### 3.2. Correlation of LncRNAs’ and DEGs’ Expression Levels

The RNA-Seq analysis revealed 14512 expressed transcripts in the hypothalamus of hypertensive ISIAH and normotensive control WAG rats. Among them, for 192 genes, there were interstrain differences in the level of transcription (Appendix A).

The correlation of expression of these DEGs with lncRNAs’ expression was calculated using the Pearson correlation coefficient. It turned out that the expression of only 12 of the 22 lncRNAs listed in Table 1 correlates with the level of DEGs’ transcription. Pearson’s coefficients of correlation between lncRNAs’ and DEGs’ expression are given in Appendix A. The number of DEGs whose expression correlates with the expression of lncRNAs in the hypothalamus of hypertensive ISIAH and normotensive WAG rats is shown in Figure 1. The expression of eight lncRNAs correlated with the expression of one or several DEGs, and a large number of correlations were found for four lncRNAs: Snhg4 correlated with the expression of 42 DEGs, LOC100910237 correlated with the expression of 93 DEGs, RGD1562890 with the expression of 29 DEGs, and Tnxa-ps1 with the expression of 112 DEGs. In total, 12 lncRNAs correlating with one or more DEGs relate to the expression of 176 (91.7%) DEGs, i.e., almost all the identified DEGs (Appendix A) implying their important role in the biological processes underlying the interstrain differences.

### 3.3. Validation of the LncRNAs’ Differential Expression

qPCR was carried out to confirm the differential transcription levels and was applied to the list of lncRNAs (Snhg4, Bc1, Gas5, LOC100910973, LOC100910237, and LOC100909675, see Table 1) until no interstrain differences were detectable. Given that the expression of two more lncRNAs (RGD1562890 and Tnxa-ps1) manifested a significant correlation with a large number of DEGs, these two lncRNAs were also included in the analysis. qPCR was performed on groups of ISIAH (*n* = 10) and WAG (*n* = 9) rats aged three months. As an internal control, we used the geometric mean of the amplification data of three reference genes: *Gapdh, Hprt*, and *Nono*. qPCR confirmed differential expression levels for 7 of the 8 analyzed lncRNAs (Figure 2). qPCR measurements did not match the RNA-Seq results for two lncRNAs (Gas5 and Tnxa-ps1), thus implying individual differences in the expression of these lncRNAs in the rat hypothalamus.

### 3.4. Functional Analysis of the DEGs Whose Expression Correlates with the Expression of LncRNAs

The results of the functional analysis of the DEGs whose expression correlated with the expression of Snhg4, LOC100910237, Tnxa-ps1 and RGD1562890 revealed numerous biological processes related to the response to various exogenous and endogenous factors, including hormonal stimuli, as well as processes participating in cell adhesion and in the functioning of the immune system (Figure 3). Lists of DEGs associated with the GO terms are presented in Appendix A.

The DEGs whose expression correlated with the expression of Snhg4, LOC100910237, Tnxa-ps1 and RGD1562890 were also used for the construction of the functional enrichment networks by means of STRING database options. This analysis helps to assess functional relations between protein products of contiguous genes.

The constructed functional enrichment network for lncRNA Snhg4 is characterized by a low protein–protein interaction (PPI) enrichment *p*-value: 0.0778 (Figure 4). This means either that the set of proteins in question is a random collection of proteins that are not very well connected or that these proteins have not been studied very much and their interactions might not yet be documented in the STRING database. Nonetheless, according to the results in Figure 4, several key nodes can be distinguished: insulin-like growth factor 2 (Igf2); insulin-like growth factor binding protein 2 (Igfbp2); vitronectin (Vtn); RT1 class II, locus Ba (RT1-Ba) and RT1 class II, and locus Bb (RT1-Bb).

Functional enrichment analysis of the lncRNA LOC100910237 network uncovered a possible association of this lncRNA with retinol metabolism, and retinol dehydrogenase activity, as well as oxidation–reduction (redox) processes (Figure 5).

Functional enrichment analysis of the lncRNA Tnxa-ps1 network revealed a possible association of this lncRNA with retinol metabolism, and retinol dehydrogenase activity, type I diabetes mellitus, antigen processing and presentation, and the metapathway associated with biotransformation, i.e., chemical alteration of chemicals such as nutrients, amino acids, toxins, and drugs in the body (Figure 6).

Functional enrichment analysis of the lncRNA RGD1562890 network uncovered a possible association of this lncRNA with both antigen processing and presentation and type 1 diabetes mellitus (Figure 7).

### 3.5. Correlating Expression Levels between LncRNAs and DEGs Associated with the Behavior/Neurological Phenotype

According to the RGD database, 18 hypothalamic DEGs were designated as associated with the behavior/neurological phenotype (Table 2). Some of them are also associated with the following terms: grooming, abnormal locomotor behavior (vertical activity and hyperactivity), abnormal cognition and abnormal emotion/affect behavior including abnormal response to novelty (Table 2). Among these 18 behavior-associated genes, two genes (*Foxg1* and *Grhl3*) code for transcription factors. Expression of 16 of the 18 DEGs related to the behavior/neurological phenotype correlated with the expression of one or more of four lncRNAs (LOC100910237, RGD1562890, Snhg4, and Tnxa-ps1; Figure 8A). The most significant correlations were found between the expression of *Abca1* and lncRNA Snhg4 (*r* = −0.975) and between the expression of *Chrnb3* and lncRNA LOC100910237 (*r* = −0.978). Expression of the *Igf2* gene correlated with the expression of all four lncRNAs.

### 3.6. Correlating Expression Levels between LncRNAs and DEGs Associated with Hypertension and Blood Pressure Regulation

In our work, 20 DEGs associated with hypertension were identified (Table 3). The expression of 17 of them correlated with the expression of one or more lncRNAs (Figure 8B). The greatest number of expression correlations was found between the DEGs associated with hypertension and LOC100910237, RGD1562890, Snhg4, and Tnxa-ps1. The most significant expression correlation was detected between *Igfbp2* and lncRNA Snhg4 (*r* = 0.975). The analysis revealed three genes (*Cyp11b2, Ptgds*, and *Slc6a3*) related to both hypertension and the behavior/neurological phenotype. *Cyp11b2* expression correlates with LOC100910237 expression, and *Ptgds* expression correlates with Snhg4 and Tnxa-ps1 expression. No correlation between *Slc6a3* expression and lncRNAs’ expression was found in our study.

## 4. Discussion

In our work, 22 lncRNAs were identified in the hypothalamic transcriptomes of hypertensive ISIAH and normotensive WAG male rats at the age of three months. Among them, seven lncRNAs (Bc1, RGD1562890, Gas5, LOC100910973, Snhg4, LOC100910237, andTnxa-ps1) were confirmed by qPCR as differentially expressed. This result is in good agreement with the previously reported observations that lncRNAs that did not reach the false discovery rate cutoff based on RNA-Seq could in fact be differentially expressed according to qPCR [10]. Furthermore, we would like to point out that the results of the assay of expression of two lncRNAs (Gas5 and Tnxa-ps1) did not match between the two methods (RNA-Seq and qPCR). This indicates the presence of individual differences in three samples in RNA-Seq analysis. In our case, of course, qPCR results should be considered a more correct fold difference because this analysis was performed on a larger number of RNA samples. Nonetheless, it should be noted that Gas5 and Tnxa-ps1 lncRNAs’ expression can vary widely, and therefore they should not be chosen as potential targets for diagnostics or therapy.

Brain cytoplasmic RNA 1 (Bc1) lncRNA was the most abundant and differentially expressed according to both methods. Bc1 is known to be involved in the translational control at the synapse. The 3′ BC1 domain specifically interacts with eukaryotic initiation factor 4A (eIF4A), thereby leading to inhibition of the rate-limiting step in the assembly of translation initiation complexes [42]. This BC1-mediated translational repression can regulate dendritic protein synthesis and have an effect on dendritic development and plasticity [43]. In our experiment, Bc1 expression did not correlate with DEGs’ expression, which is in good agreement with the above finding that this lncRNA takes part in the control over translation.

In behavioral testing of Bc1 RNA–deficient mice, decreased exploratory activity and increased anxiety have been documented [44]. Our experiment showed upregulation of *Bc1* in the hypothalamus of ISIAH rats, which are characterized by an increase in locomotor and exploratory activity in an unfamiliar environment [32]. Accordingly, our data are well consistent with the notion of an association between the level of Bc1 expression and behavior.

In the present study, the greatest number of expression correlations were found between DEGs and lncRNAs RGD1562890, Snhg4, LOC100910237, and Tnxa-ps1.

Expression of lncRNA RGD1562890 in the brain of Fischer 344 rats has been confirmed [45], but to date, this lncRNA has been poorly characterized functionally. In cultured hippocampal neurons, RGD1562890 is reported to be associated with RNA-binding protein hnRNP K (heterogeneous nuclear ribonucleoprotein K), which is present at the synapse and is modulated by the neurotrophin BDNF (brain-derived neurotrophic factor) [46]. Heterogeneous nuclear ribonucleoprotein can mediate insulin-driven inhibition of renal angiotensinogen gene expression and prevent hypertension and kidney injury in diabetic mice [47]. Our experiment shows for the first time that lncRNA RGD1562890 is expressed in the hypothalamus of rats. Functional enrichment analysis indicated that six of the 29 DEGs whose expression correlated with lncRNA RGD1562890 expression are associated with type 1 diabetes mellitus. GO analysis uncovered a correlation of RGD1562890 expression with DEGs associated with various biological processes, among which “immune system processes” and “response to stimulus including response to glucocorticoid” deserve the most attention from our point of view. On the basis of the above data of other authors [46,47] and on the results of our study, we can assume that the differential expression of lncRNA RGD1562890 may help to regulate the manifestation of some features of the hypertensive phenotype in ISIAH rats.

Snhg4 (small nucleolar RNA host gene 4) belongs to the SNHG family of 22 members. SNHGs participate in the initiation and progression of various endocrine-system–related cancers and other diseases [48,49]. SNHGs are present in both the cytoplasm and nucleus. Five main types of molecular mechanisms of their action have been described: an influence on DNA methylation, regulation of transcription, suppression of translation, sponging of microRNAs (miRNAs), and stabilization of proteins. Nevertheless, SNHG4 mainly acts as a sponge for miRNAs, i.e., SNHG4 serves as a competing endogenous RNA by forming a lncRNA–miRNA–mRNA regulatory network [49]. Numerous studies have shown that SNHG4 is upregulated in various cancers, where its activity is implicated in cell proliferation, invasiveness, migration, and inhibition of apoptosis (reviewed in [49]). SNHG4 has also been shown to have the anti-inflammatory effect in microglia after cerebral ischemia–reperfusion injury [50]. In studies on hypertensive rat models, a decrease in Snhg4 expression in the renal cortex has been demonstrated in spontaneously hypertensive rats compared with control WKY rats [12]. Downregulation of Snhg4 expression in the hypothalamus of hypertensive rats was shown for the first time in the present study.

The function of tenascin XA, pseudogene 1 (Tnxa-ps1) has also been investigated previously. A knockdown of this lncRNA (Tnxa-ps1) promotes neuronal survival by inhibiting apoptosis in primary culture of hippocampal neurons under oxygen–glucose deprivation/reperfusion [51]. Expression of Tnxa-ps1 has also previously been associated with the manifestation of a hypertensive phenotype. The level of lncRNA Tnxa-ps1 (formerly known as NR024118) is remarkably higher in the myocardium of spontaneously hypertensive rats [52]. Angiotensin II reduces the lncRNA Tnxa-ps1 (NR024118) amount in rat cardiac fibroblasts [53], and this effect is mediated by angiotensin II type 1 receptor–dependent signaling [54]. Tnxa-ps1 can play a regulatory part by acting as a sponge of miRNAs [55].

LncRNA LOC100910237 is functionally uncharacterized to date, although its expression in the Fischer 344 rat brain has been confirmed [45]. Our work shows for the first time the upregulation of lncRNA LOC100910237 in the hypothalamus of hypertensive ISIAH rats compared to normotensive WAG rats. In our project, for this lncRNA, correlations with the level of expression of 93 DEGs were found. The calculation of the Pearson coefficient of correlation between lncRNA and mRNA expression levels is regarded by many authors as a key step in the identification of coexpressed lncRNA–mRNA pairs [17,18,56]. Taking into account that lncRNA LOC100910237 expression correlated here with many DEGs associated with both the behavior/neurological phenotype and hypertension, we propose that the expression features of this lncRNA may be connected to the development of phenotypic characteristics of the ISIAH rat strain.

Significantly altered levels of expression of several lncRNAs in the hypothalamus of ISIAH rats imply their participation in the formation of the phenotypic features of ISIAH rats, which were created via selection for a sharp increase in blood pressure under conditions of short-term restraint stress. Such stress can be considered mild emotional stress, which nevertheless led to the selection of the genotype characterized by spontaneously developing high blood pressure, high stress reactivity in a changing environment [27,28], and specific animal behavior [32].

The question of the possibility of joint genetic control of such traits as behavior and hypertension has been relevant for many years and attracts the attention of many researchers. The existence of such a possibility is in good agreement with data in the RGD database, which contains 789 genes associated with hypertension, of which 376 genes are associated with both hypertension and a behavior/neurological phenotype (as of 24 March 2022). Our results suggest that lncRNAs can contribute to the coregulation of these traits.

In our comparative analysis of hypothalamic gene expression profiles between ISIAH and WAG rats, 18 DEGs related to the behavior/neurological phenotype were identified. Among them, three DEGs (*Cyp11b2, Ptgds*, and *Slc6a3*) were found to be associated with both behavior and hypertension.

*Cyp11b2* (cytochrome P450, family 11, subfamily b, polypeptide 2) encoding aldosterone synthase is known as a key player in steroidogenic pathways because it takes part in aldosterone biosynthesis [57]. Aldosterone has been reported to influence both behavioral (mood, appetite, and exploratory behavior) and autonomic (baroreceptor) functions [58]. Aldosterone synthesized in the brain serves as a local ligand for autocrine or paracrine activation of mineralocorticoid receptors [59], which enhances salt appetite, sympathetic drive, and a vasopressin release, promoting the development of salt-sensitive hypertension [60,61].

The upregulation of *Cyp11b2* in the hypothalamus of ISIAH rats, which has previously been confirmed by qPCR [33], suggests that the excessive aldosterone synthesis in the hypothalamus of ISIAH rats may contribute to the pathogenesis of stress-sensitive hypertension and to the observed changes in behavior linked with increased anxiety in ISIAH rats. Because, as we demonstrated, *Cyp11b2* expression correlates with LOC100910237 expression, we can theorize that lncRNA LOC100910237 is connected to these processes.

*Ptgds* (prostaglandin D2 synthase, L-PGDS) is responsible for the biosynthesis of prostaglandin D2 (PGD2), which is the most abundant prostaglandin in the brain. L-PGDS enables retinoid-binding activity [62]. It may be activated in response to a glucocorticoid [63]. PGD2 is implicated in the modulation of inflammation [64]. Its G_α__(s)_ protein–coupled receptor (DP1) protects brain from ischemia–reperfusion injury [65]. L-PGDS, via production of PGD2, plays a substantial role in the glucose intolerance associated with type 2 diabetes mellitus [66] and is associated with attention deficit hyperactivity disorder [67]. L-PGDS deletion reduces spontaneous locomotor activity [68]. The absence or inhibition of L-PGDS results in dyslipidemia and altered expression of lipogenesis genes [69]. PGD2 is also linked with functional properties of vascular smooth muscle cells [70]. According to our study, *Ptgds* expression correlates with Snhg4 and Tnxa-ps1 expression. Thus, we can hypothesize a function of lncRNAs Snhg4 and Tnxa-ps1 in the regulation of processes related to the expression of *Ptgds*, which may be associated with the manifestation of specific features of the hypertensive phenotype of ISIAH rats.

*Slc6a3* [solute carrier family 6 (neurotransmitter transporter, dopamine), member 3] encodes a protein (DAT) that transports the released dopamine from the synaptic cleft into presynaptic terminals of the brain, thereby terminating dopaminergic neurotransmission. It has been demonstrated that, in knockout (DAT^−/^^−^) animals, the concentration of extracellular dopamine significantly increases and open field activity is also enhanced [71]. Accordingly, the significantly weaker *Slc6a3* transcription in the hypothalamus of ISIAH rats compared to WAG rats is suggestive of enhanced dopaminergic transmission in the hypothalamus of hypertensive ISIAH rats; this phenomenon may explain the observed high locomotor activity of ISIAH rats in the open field test. In our paper, the expression of *Slc6a3* does not correlate with the expression of lncRNAs; however, such correlations were found by us for other DEGs related to the modulation of dopaminergic synaptic signaling, which are described in more detail below.

The expression of lncRNA LOC100910237 correlated with the expression of two genes (*Chrnb3* and *Grm2*) associated with neurotransmission processes, including dopaminergic synaptic signaling.

*Chrnb3* (cholinergic receptor, nicotinic, beta 3) is thought to be involved in various processes, including acetylcholine receptor activity. Analysis of mice lacking the beta 3 nicotinic receptor subunit has revealed that a population of nicotinic acetylcholine receptors containing the β3 nicotinic receptor subunit modulates the striatal dopamine release and alters behavior [72]. Mice with a knockout of the β3 nicotinic receptor subunit demonstrate significantly higher levels of locomotor activity in the open field arena and less pronounced anxiety-like behavior [72,73].

*Grm2* (glutamate receptor, metabotropic 2) encodes a presynaptic metabotropic glutamate receptor, mGlu2. In the hypothalamus, glutamate and its receptors are crucial for neuroendocrine regulation of hormone secretion [74]. Metabotropic glutamate receptors have been categorized into three groups on the basis of sequence homology, putative signal transduction mechanisms, and pharmacological properties. Group II metabotropic glutamate receptors (mGlu2 and mGlu3, encoded by *Grm2* and *Grm3*) affect striatal dopamine, and the effect may contribute to the behavioral phenotype as shown in research on double-knockout mice lacking mGlu2 and mGlu3 (*mGlu2/3*^−^^/^^−^) [75]. The reduced mGlu2 activity may protect from stress-induced changes underlying the onset or recurrence of stress-induced disorders [76].

Dopamine is a neurotransmitter produced in various brain structures, including the hypothalamus. Dopamine can modulate anxiety-like behavior [77], and dopamine dysfunction has been implicated in various nervous-system diseases [78]. In addition, an abnormal dopamine system contributes to the development of hypertension in spontaneously hypertensive rats, which exhibit behavioral traits featuring the main symptoms of attention deficit hyperactivity disorder of humans [79,80]. Our previous investigation indicates that the level of dopamine is elevated in the hypothalamus of 12-week-old ISIAH rats [28]. Based on the foregoing, it can be hypothesized that lncRNA LOC100910237 helps to govern the hypothalamic processes connected to dopaminergic synaptic signaling, which may be implicated in the formation of the hypertensive phenotype in ISIAH rats, including their behavioral features.

Experimental findings that two other lncRNAs (Snhg4 and Tnxa-ps1) are involved in the regulation of processes related to dopaminergic signaling have not yet been reported. Nonetheless, in this work, we found indirect evidence that this is possible. Namely, expression levels of Snhg4 and Tnxa-ps1 correlate with the transcription level of *Abca1* [ATP-binding cassette, subfamily A (ABCA), member 1]. *Abca1* encodes a membrane-associated protein known as a cholesterol transporter. Some studies indicate that the cholesterol content of brain membranes is linked with DAT-mediated dopamine neurotransmission [81]. Recently, cholesterol transporter ABCA1 was reported to be involved in the control over DAT activity [82]. It should be emphasized that the expression correlation between Snhg4 and Abca1 is one of the most statistically significant in our study.

Another statistically highly significant correlation was detected between Snhg4 and *Igfbp2* expression (insulin-like growth factor-binding protein 2). According to the functional enrichment analysis of the lncRNA Snhg4 network (Figure 3), the protein encoded by *Igfbp2* may be considered one of the key nodes. IGFBP2, through binding to IGF, can influence such IGF functions as development and growth. The level of *Igfbp2* transcription proved to be low in the hypothalamus of ISIAH rats here. Low serum IGFBP2 concentration is associated with a higher risk of obesity-related insulin resistance [83,84]. Inhibition of IGFBP2 is also related to immunosuppression [85]. It was recently suggested that IGFBP2 is also involved in the regulation of the neuronal plasticity that modulates spatial learning and memory [85]. It has recently been suggested that IGFBP2 is also involved in the regulation of neuronal plasticity, which modulates spatial learning and memory [86].

Functional enrichment analysis of the lncRNAs LOC100910237 and Tnxa-ps1 networks (Figure 4 and Figure 5) uncovered for the first time a possible association of these lncRNAs with retinol metabolism, and retinol dehydrogenase activity. Although the role of the retinol metabolic pathway in the development of hypertension is not fully understood at present, there is some evidence that retinol metabolism may be an important link in the complex pattern of metabolic changes associated with the pathogenesis of hypertension [87,88]. Our data are in a good agreement with this point of view and indicate that lncRNAs LOC100910237 and Tnxa-ps1 may participate in this process in the hypothalamus of ISIAH rats simulating the stress-sensitive form of arterial hypertension.

## 5. Conclusions

Here, expressed lncRNAs were identified in the hypothalamus of hypertensive ISIAH and normotensive WAG male rats. Seven of these lncRNAs showed interstrain differences in expression levels. Four lncRNAs were identified (RGD1562890, Snhg4, LOC100910237, and Tnxa-ps1) whose expression correlated with the level of transcription of many DEGs. Functional annotation of the DEGs whose expression correlated with the level of the lncRNAs’ expression revealed groups of genes associated with immune-system processes and with responses to various exogenous and endogenous factors, including hormonal stimuli, the most significant of which seem to be the responses to glucocorticoid hormones, to estradiol, and to retinoic acid. We detected several lncRNAs in the hypothalamus that may participate in the formation of the hypertensive state of ISIAH male rats, which are characterized by high stress reactivity under the conditions of emotional stress and special features of behavior in an unfamiliar environment. Of course, it should be emphasized that, in this work, not all, but only lncRNAs with a polyA tail were analyzed and discussed because of the mRNA isolation technology used for RNA-Seq. Nevertheless, the chosen analytical conditions allowed us to obtain new data and for the first time to show hypothalamic differential expression between hypertensive and normotensive rats for several lncRNAs and to discuss in detail their possible involvement in biological processes that may be connected to the development of the hypertensive phenotype.

## Figures and Tables

**Figure 1 genes-13-01598-f001:**
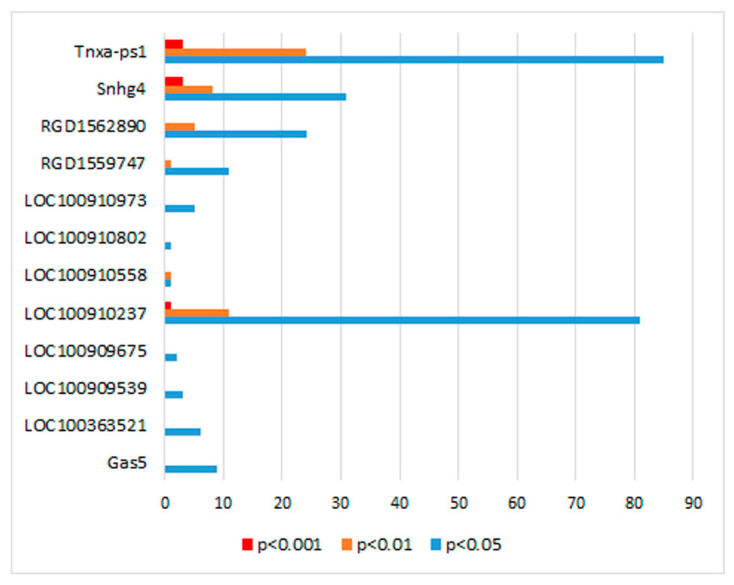
The number of DEGs whose expression correlates with the expression of lncRNAs in the hypothalamus of hypertensive ISIAH and normotensive WAG rats. Correlations with the level of transcription of genes differentially expressed in the hypothalamus of ISIAH and WAG rats were found only for 12 out of the 22 expressed lncRNAs listed in Table 1. The color indicates the significance of the correlation. The levels of significance are based on the two-tailed test (Pearson’s correlation analysis). The *x*-axis shows the number of DEGs.

**Figure 2 genes-13-01598-f002:**
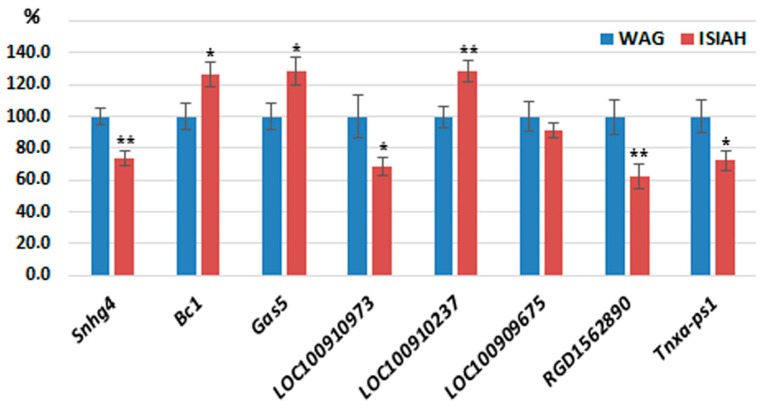
Relative abundance of lncRNAs measured by qPCR in the hypothalamus of ISIAH and WAG rats. The normalized RNA level in control samples of the WAG rats was set to 100%. Vertical bars denote standard error of the mean. ISIAH (*n* = 10), WAG (*n* = 9). As an internal control, we employed the geometric mean of the amplification data of three reference genes: *Gapdh*, *Hprt*, and *Nono*. * *p* < 0.05; ** *p* < 0.01 (Student’s *t*-test).

**Figure 3 genes-13-01598-f003:**
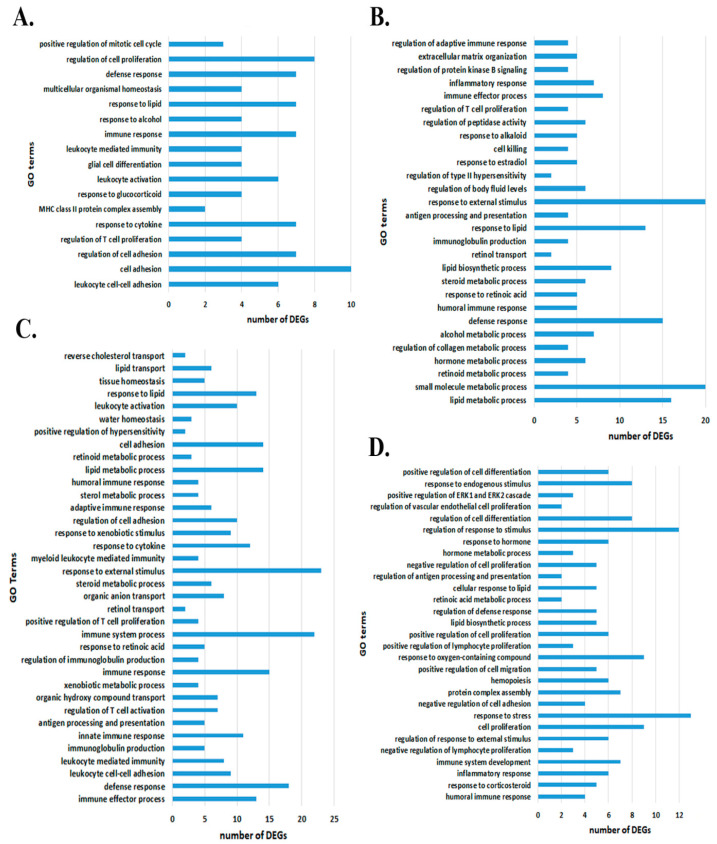
GO terms for DEGs correlating with lncRNAs: (**A**) Snhg4; (**B**) LOC100910237; (**C**) Tnxa-ps1; (**D**) RGD1562890. Significantly (*p* < 0.05) enriched biological processes are shown.

**Figure 4 genes-13-01598-f004:**
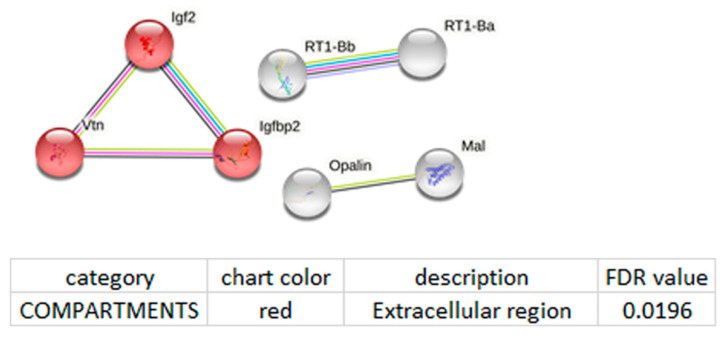
Functional enrichment analysis of the lncRNA Snhg4 network. Functional enrichment network was constructed by means of the STRING database [22] using the DEGs whose expression correlated with the expression of lncRNA Snhg4. Each node represents all the proteins produced by a single protein-coding gene. The key nodes are insulin-like growth factor 2 (Igf2); insulin-like growth factor binding protein 2 (Igfbp2); vitronectin (Vtn); RT1 class II, locus Ba (RT1-Ba) and RT1 class II, locus Bb (RT1-Bb). Edges represent protein–protein associations. Purple lines indicate experimentally determined interactions; blue lines denote known interactions from curated databases; black lines indicate coexpression; green lines mean results of text mining; and grey lines indicate protein homology. The protein–protein interaction (PPI) enrichment *p*-value is 0.0778; this means either that the current set of proteins is a random collection of proteins that are not very well connected or that these proteins have not been studied very much and their interactions might not yet be documented in STRING.

**Figure 5 genes-13-01598-f005:**
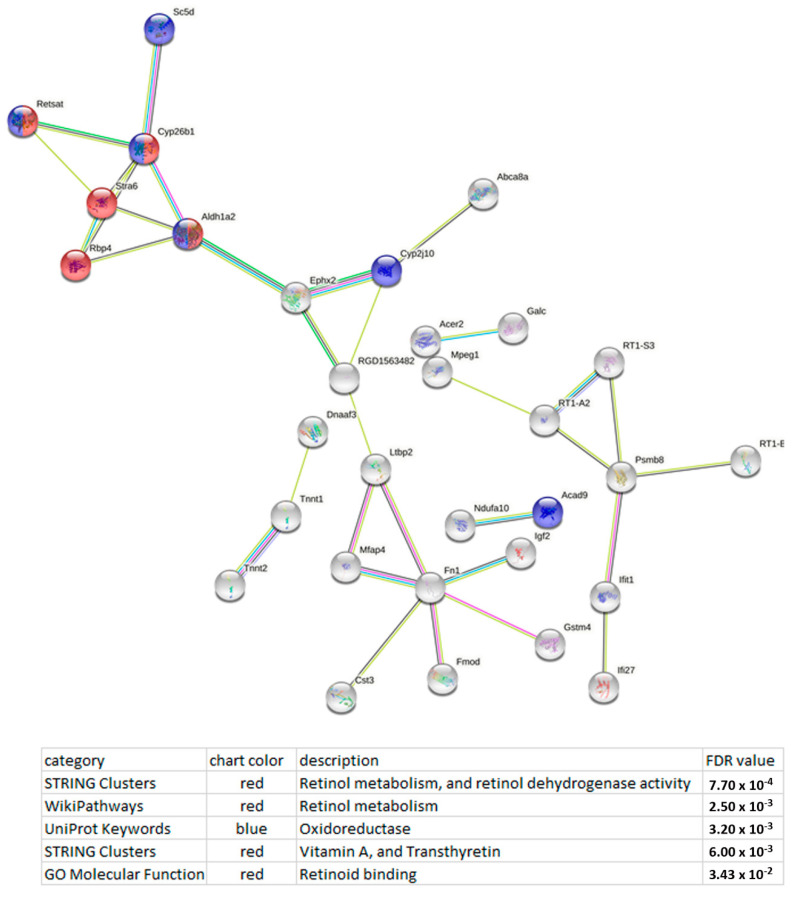
Functional enrichment analysis of the lncRNA LOC100910237 network points to possible association of this lncRNA with retinol metabolism and oxidation–reduction (redox) processes. The functional enrichment network was constructed with the help of the STRING database [22] using the DEGs whose expression correlated with the expression of lncRNA LOC100910237. Each node represents all the proteins produced by a single protein-coding gene. The key nodes are retinol binding protein 4 (Rbp4); cytochrome P450, family 26, subfamily b, polypeptide 1 (Cyp26b1); retinol saturase (Retsat); signaling receptor and transporter of retinol STRA6 (Stra6); and aldehyde dehydrogenase 1 family, member A2 (Aldh1a2). Purple lines indicate experimentally determined interactions; blue lines represent known interactions from curated databases; dark blue lines denote gene co-occurrence; black lines indicate coexpression; green lines mean results of text mining; and grey lines indicate protein homology. Protein–protein interaction (PPI) enrichment *p*-value: 3.02 × 10^−5^.

**Figure 6 genes-13-01598-f006:**
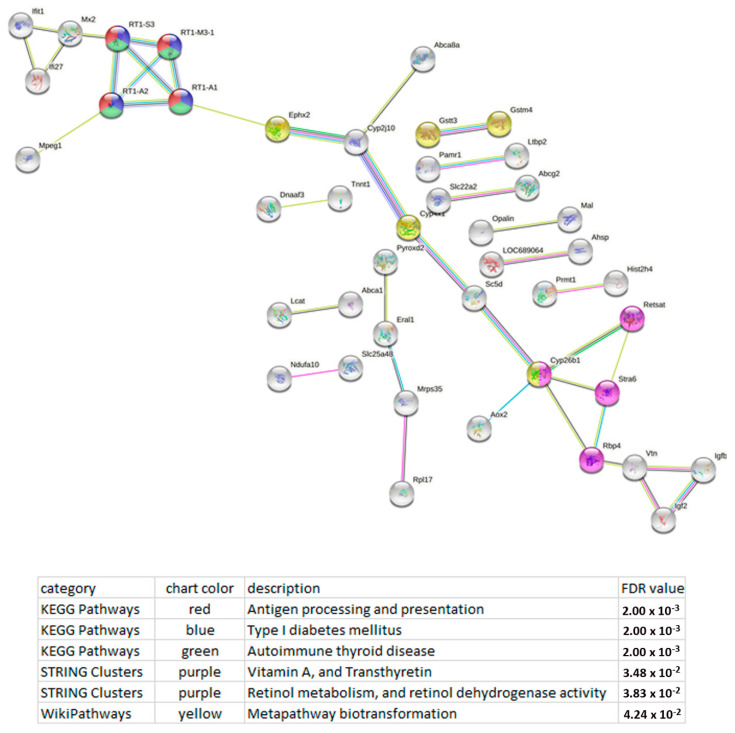
Functional enrichment analysis of the lncRNA Tnxa-ps1 network points to possible association of this lncRNA with several pathways: retinol metabolism, antigen processing and presentation, type 1 diabetes mellitus and metapathway biotransformation. The functional enrichment network was constructed by means of the STRING database [22] using the DEGs whose expression correlated with the expression of lncRNA Tnxa-ps1. Each node represents all the proteins produced by a single protein-coding gene. The key links associated with retinol metabolism are retinol binding protein 4 (Rbp4); cytochrome P450, family 26, subfamily b, polypeptide 1 (Cyp26b1); retinol saturase (Retsat); and signaling receptor and transporter of retinol STRA6 (Stra6). The key links associated with antigen processing and presentation and type 1 diabetes mellitus are RT1 class Ia, locus A1 (RT1-A1); RT1 class Ia, locus A2 (RT1-A2); RT1 class II, locus Bb (RT1-Bb); RT1 class Ib, locus M3, gene 1 (RT1-M3-1); and RT1 class Ib, locus S3 (RT1-S3). The key links associated with metapathway biotransformation are epoxide hydrolase 2 (Ephx2); cytochrome P450, family 4, subfamily x, polypeptide 1 (Cyp4x1); cytochrome P450, family 26, subfamily b, polypeptide 1 (Cyp26b1); glutathione S-transferase, theta 3 (Gstt3); and glutathione S-transferase mu 4 (Gstm4). Edges represent protein–protein associations. Purple lines indicate experimentally determined interactions; blue lines mean known interactions from curated databases; dark blue lines denote gene co-occurrence; black lines indicate coexpression; green lines denote results of text mining; and grey lines indicate protein homology. Protein–protein interaction (PPI) enrichment *p*-value: 9.75 × 10^−6^.

**Figure 7 genes-13-01598-f007:**
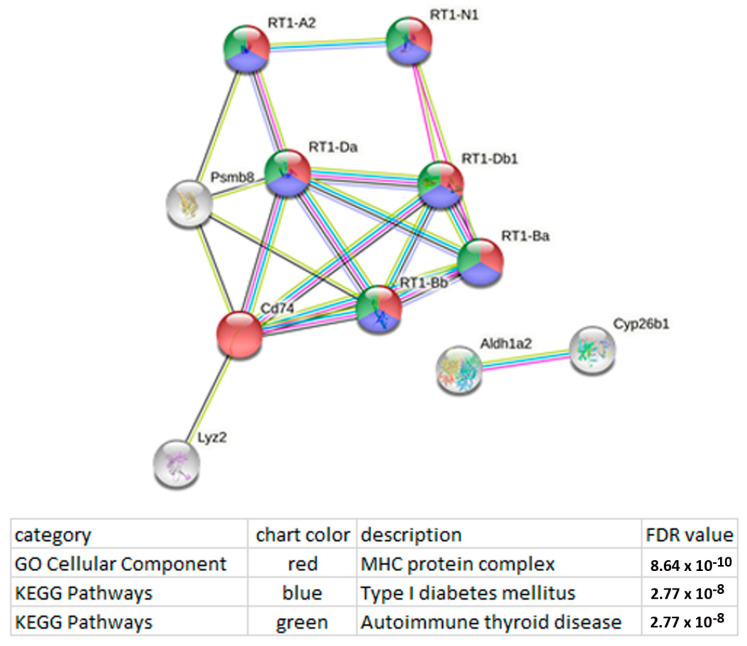
Functional enrichment analysis of the lncRNA RGD1562890 network points to possible participation of this lncRNA in both antigen processing and presentation and type 1 diabetes mellitus. The functional enrichment network was constructed by means of the STRING database [22] using the DEGs whose expression correlated with the expression of lncRNA RGD1562890. Each node represents all the proteins produced by a single protein-coding gene. The key links associated with antigen processing and presentation and type 1 diabetes mellitus are Cd74 (CD74 molecule); Lyz2 (lysozyme 2); Psmb8 (proteasome 20 S subunit beta 8); RT1-A2 (RT1 class Ia, locus A2); RT1-Ba (RT1 class II, locus Ba); RT1-Bb (RT1 class II, locus Bb); RT1-Da (RT1 class II, locus Da); RT1-Db1 (RT1 class II, locus Db1); RT1-N2 (RT1 class Ib, locus N2). Edges represent protein–protein associations. Purple lines indicate experimentally determined interactions; blue lines denote known interactions from curated databases; dark blue lines represent gene co-occurrence; black lines indicate coexpression; green lines represent results of text mining. Protein–protein interaction (PPI) enrichment *p*-value: 1.11 × 10^−15^.

**Figure 8 genes-13-01598-f008:**
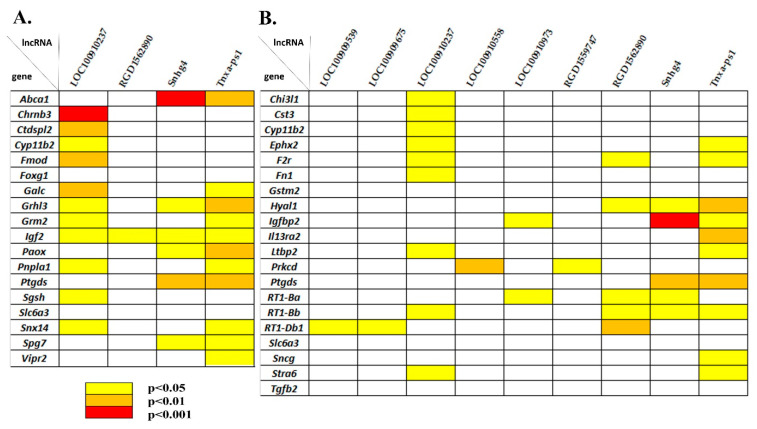
Correlation analysis of levels of expression between lncRNAs and DEGs associated with (**A**) the behavior/neurological phenotype or (**B**) hypertension. The greatest number of correlations was found between differentially expressed lncRNAs Snhg4, LOC100910237, RGD1562890 and Tnxa-ps1 and DEGs related to both the behavior/neurological phenotype and hypertension. The color indicates the levels of significance based on the two-tailed test (Pearson’s correlation analysis).

**Table 1 genes-13-01598-t001:** The expression level of lncRNAs detected in the hypothalamus of ISIAH and WAG rats (RNA-Seq data).

Gene ID	LncRNA	Chr.:Megabases	Expression Level in		log_2_ (Fold Change) of ISIAH/WAG	*p*-Value
			ISIAH, RPKM	WAG, RPKM		
100568448	Snhg4 *	18:28.07	13.7	40.6	−1.56	0.0057
29294	Bc1	1:224.94	127,304.0	91,730.1	0.47	0.0128
81714	Gas5 *	13:83.75	15.3	21.4	−0.49	0.0409
100910973	LOC100910973 *	1:234.18	24.5	30.9	−0.34	0.0676
100910237	LOC100910237 *	1:24.23	6.1	4.1	0.59	0.0701
100909675	LOC100909675 *	4:7.83	1.5	1.2	0.29	0.3363
100363521	LOC100363521 *	4:43.75	31.7	34.9	−0.14	0.4049
294945	RGD1562890 *	2:124.12	2.8	3.6	−0.37	0.4079
100910802	LOC100910802 *	12:24.79	1.9	1.5	0.32	0.4600
29536	Rmrp	15:31.57	20.0	16.4	0.28	0.4703
312310	RGD1559747 *	4:142.93	7.6	7.0	0.12	0.4978
100909539	LOC100909539 *	2:105.05	12.2	13.7	−0.16	0.5426
25602	Tnxa-ps1 *	20:6.40	8.1	4.5	0.83	0.7158
680254	LOC680254	13:79.98	5.7	5.4	0.08	0.7843
309109	Tmem80	1:221.20	17.2	16.6	0.06	0.7931
100910558	LOC100910558 *	5:104.68	2.2	2.1	0.03	0.8448
29206	Ybx1-ps3	10:40.22	23.4	23.8	−0.03	0.8541
500568	RGD1564482	5:158.41	1.3	1.4	−0.15	0.8674
259227	Vof16	8:44.27	3.0	3.1	−0.04	0.8873
360865	Prrc2c	13:85.44	29.5	29.1	0.02	0.8921
686120	LOC686120	12:26.72	1.4	1.4	0.01	0.9253
100911177	LOC100911177	3:169.95	89.9	89.7	0.00	0.9913

* lncRNAs correlating with DEG expression. RPKM (reads per kilobase per million mapped reads) represents the average measured level of expression in three samples in each group of rats analyzed by RNA-Seq. A positive log_2_ (fold change) of ISIAH/WAG indicates a higher level of measured transcription in the hypothalamus of ISIAH rats. Negative log_2_(fold change) of ISIAH/WAG denotes a lower level of measured transcription in the hypothalamus of ISIAH rats.

**Table 2 genes-13-01598-t002:** Hypothalamic DEGs associated with the behavior/neurological phenotype.

Gene	Chr.:Megabases	Expression Level in		log_2_ (Fold Change) of ISIAH/WAG	*q*-Value	Definition
		WAG, RPKM	ISIAH, RPKM			
*Abca1* ^∆^	5:74.0	7.47	11.71	0.6	5.94 × 10^−3^	ATP-binding cassette subfamily A member 1
*Chrnb3* ^∆¶^	16:68.5	3.12	1.31	−1.2	5.94 × 10^−3^	cholinergic receptor nicotinic beta 3 subunit
*Ctdspl2*	3:120.5	4.97	7.81	0.7	2.47 × 10^−2^	CTD small phosphatase-like 2
*Cyp11b2* ^¥^	7:116.2	0.90	2.07	1.2	1.52 × 10^−2^	cytochrome P450, family 11, subfamily b, polypeptide 2
*Fmod* ^∆^	13: 55.9	22.13	12.70	−0.8	5.94 × 10^−3^	fibromodulin
*Foxg1* ^∆¶€^	6:79.5	5.45	2.94	−0.9	5.94 × 10^−3^	forkhead box G1
*Galc* ^∆¶^	6:131.4	10.27	6.30	−0.7	1.52 × 10^−2^	galactosylceramidase
*Grhl3 ** ^∆€^	5:157.7	1.05	2.68	1.4	5.94 × 10^−3^	grainyhead-like transcription factor 3
*Grm2* ^©∆¶^	8:114.7	10.93	2.73	−2.0	5.94 × 10^−3^	glutamate metabotropic receptor 2
*Igf2* ^∆^	1:222.7	201.43	94.08	−1.1	5.94 × 10^−3^	insulin-like growth factor 2
*Paox* ^∆¶^	1:219.5	7.27	13.01	0.8	5.94 × 10^−3^	polyamine oxidase
*Pnpla1* ^∆^	20:8.4	7.33	1.12	−2.7	5.94 × 10^−3^	patatin-like phospholipase domain-containing 1
*Ptgds* ^¥^	3:2.7	3548.34	2356.79	−0.6	1.09 × 10^−2^	prostaglandin D2 synthase
*Sgsh*	10:108.1	2.49	4.90	1.0	1.99 × 10^−2^	N-sulfoglucosamine sulfohydrolase
*Slc6a3 ** ^©§#∆¤¶¥^	1:33.7	6.86	2.65	−1.4	5.94 × 10^−3^	solute carrier family 6 member 3
*Snx14* ^©§#^	8:95.5	21.12	11.94	−0.8	5.94 × 10^−3^	sorting nexin 14
*Spg7* ^∆^	19:66.6	17.80	26.18	0.6	2.83 × 10^−2^	SPG7 matrix AAA peptidase subunit, paraplegin
*Vipr2* ^∆^	6:152.9	1.68	2.92	0.8	4.38 × 10^−2^	vasoactive intestinal peptide receptor 2

DEGs associated with * abnormal grooming in mice, ^©^ anormal emotion/affect behavior, ^§^ abnormal response to novelty, ^#^ abnormal cognition, ^∆^ abnormal locomotor behavior, ^¤^ abnormal vertical activity, ^¶^ hyperactivity, ^¥^ hypertension, and ^€^ transcription factor genes. Genes were considered differentially expressed at a false discovery rate (*q*-value) of <5%.

**Table 3 genes-13-01598-t003:** Hypothalamic DEGs associated with hypertension.

Gene	Chr.:Megabases	Expression Level in		log_2_ (Fold Change) of ISIAH/WAG	*q*-Value	Definition
		WAG, RPKM	ISIAH, RPKM			
*Chi3l1*	13:56.1	48.06	19.85	−1.3	5.94 × 10^−3^	chitinase 3-like 1
*Cst3*	3:149.6	3929.44	2609.54	−0.6	1.09 × 10^−2^	cystatin C
*Cyp11b2 **	7:116.2	0.90	2.07	1.2	1.52 × 10^−2^	cytochrome P450, family 11, subfamily b, polypeptide 2
*Ephx2*	15:48.8	0.63	13.56	4.4	5.94 × 10^−3^	epoxide hydrolase 2
*F2r*	2:45.3	12.80	18.83	0.6	3.53 × 10^−2^	coagulation factor II (thrombin) receptor
*Fn1*	9:78.7	24.70	12.54	−1.0	5.94 × 10^−3^	fibronectin 1
*Gstm2*	2:230.2	30.69	18.40	−0.7	1.09 × 10^−2^	glutathione S-transferase mu 2
*Hyal1*	8:115.7	8.44	13.64	0.7	3.79 × 10^−2^	hyaluronidase 1
*Igfbp2*	9:79.9	95.58	49.64	−0.9	5.94 × 10^−3^	insulin-like growth factor-binding protein 2
*Il13ra2*	X:118.6	0.56	1.49	1.4	2.83 × 10^−2^	interleukin 13 receptor subunit alpha 2
*Ltbp2*	6:116.9	2.65	0.36	−2.9	5.94 × 10^−3^	latent transforming growth factor beta-binding protein 2
*Prkcd*	16:6.6	9.96	15.32	0.6	4.98 × 10^−2^	protein kinase C, delta
*Ptgds **	3:2.7	3548.34	2356.79	−0.6	1.09 × 10^−2^	prostaglandin D2 synthase
*RT1-Ba*	20:6.1	9.75	5.48	−0.8	4.72 × 10^−2^	RT1 class II, locus Ba
*RT1-Bb*	20:6.1	6.56	1.69	−2.0	5.94 × 10^−3^	RT1 class II, locus Bb
*RT1-Db1*	20:6.2	11.30	6.65	−0.8	5.94 × 10^−3^	RT1 class II, locus Db1
*Slc6a3 **	1:33.7	6.86	2.65	−1.4	5.94 × 10^−3^	solute carrier family 6 member 3
*Sncg*	16:9.1	64.11	35.35	−0.9	5.94 × 10^−3^	synuclein, gamma
*Stra6*	8:62.7	8.73	14.19	0.7	5.94 × 10^−3^	signaling receptor and transporter of retinol STRA6
*Tgfb2*	13:109.7	6.73	12.03	0.8	5.94 × 10^−3^	transforming growth factor beta 2

* DEGs associated with the behavior/neurological phenotype. Genes were considered differentially expressed at a false discovery rate (*q* value) of <5%.

## Data Availability

The RNA-Seq datasets are available in the NCBI Short Read Archive database under accession number PRJNA299102.

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
