# Peer review of "Identification of Hypothalamic Long Noncoding RNAs Associated with Hypertension and the Behavior/Neurological Phenotype of Hypertensive ISIAH Rats"

_genes, 2022, doi:10.3390/genes13091598_

Round 1

Reviewer 1 Report

The manuscript by Fedoseeva et. al. studied the transcriptome of hypertensive ISIAH rats and wild type controls to identify dysregulated lncRNAs. The RNA-seq for n=3 (3 controls) rats were performed in an earlier study (2016) and the current study re-used the data to look for lncRNAs. The authors identified 22 known lncRNAs with a 5% FDR to be differentially expressed. They then went ahead to determine the possible function of these lncRNAs using correlation analysis followed by an attempt to validate these using RT-PCR. The PCR was performed on a separate group of ISIAH (n=10) and WAG (n = 9) rats. The authors could validate 7 lncRNA candidates. The manuscript is well written but lacks significant results. Here are some of my concerns:

·      The paper is good example of re-mining of published datasets. It would be nice to re-do the analysis using latest tools and not rely on results from tools which are more than a decade old. Also, some of the databases have been updated including the reference which may have resulted in new genomic annotations.

·      The idea behind the study is good but the data is from 3 vs 3 rats, which makes it hard to be generalizable. The authors could even look at public resources to increase the number or try specific lncRNA enrichment techniques.

·      Only three lncRNAs were statistically significant at p<0.05, yet the authors chose top 6 and two other based on correlation analysis for qPCR. It is not clear how the candidates were chosen? If fold-change was used as criteria, some of them show no fold-change difference between the groups, which makes them bad candidates for validation.

·      The qPCR measurements did not match the RNA-seq results for Tnxa-ps1, then why was it further chosen for correlation-based analysis? Also, it is not clear what happened to the other lncRNAs candidates? How do the authors account for randomness in the correlation analysis?

·      The network analysis performed by the authors for 4 of the lncRNA co-expressed genes shows highly non-connected genes. The authors highlighted some networks and functions in the figures. Are these the top functions from String or chosen at random by the Authors? The enrichment analysis should be performed against the identified genes as background as against all genes (default).

Minor:

·      Reduce the title to “Identification of Long Noncoding RNAs Associated with Hyper-tension and the Behavior/Neurological Phenotype of Hypertensive ISIAH Rats”

·      It’s not clear from the abstract or text that the study is a validation of earlier study and not done again.

·      Please mention version numbers of genome annotations used.

·      String figures can be re-done using only connected nodes (and use of tools like Cytoscape)

Author Response

Comments and Suggestions for Authors

The manuscript by Fedoseeva et. al. studied the transcriptome of hypertensive ISIAH rats and wild type controls to identify dysregulated lncRNAs. The RNA-seq for n=3 (3 controls) rats were performed in an earlier study (2016) and the current study re-used the data to look for lncRNAs. The authors identified 22 known lncRNAs with a 5% FDR to be differentially expressed. They then went ahead to determine the possible function of these lncRNAs using correlation analysis followed by an attempt to validate these using RT-PCR. The PCR was performed on a separate group of ISIAH (n=10) and WAG (n = 9) rats. The authors could validate 7 lncRNA candidates. The manuscript is well written but lacks significant results. Here are some of my concerns:

  • The paper is good example of re-mining of published datasets. It would be nice to re-do the analysis using latest tools and not rely on results from tools which are more than a decade old. Also, some of the databases have been updated including the reference which may have resulted in new genomic annotations.

Answer: Yes, you are right, our work is indeed a continuation of previously published data. Since then, genome annotations have been significantly updated and the number of transcribed non-coding RNAs has greatly increased. Annotating our data with the rn7.0 version significantly increases the number of non-coding RNAs. This is certainly great progress. However, annotation of non-coding RNAs in different databases (NCBI and Ensembl) gives very different results. NCBI and ENSEMBL have now launched a joint MANE annotation aggregation initiative https://www.ncbi.nlm.nih.gov/refseq/MANE/ but for now only for humans. Therefore, we decided not to rebuild the article for the new lists of lncRNAs, but just rechecked the annotations of those presented in Table 1 of the manuscript.

We dare to hope that the current version of the manuscript will be of interest to readers, since the manuscript presents original results that for the first time identified seven lncRNAs differentially expressed in the hypothalamus of hypertensive ISIAH and normotensive WAG rats.  Among them, two lncRNAs (LOC100910237 and RGD1562890) have not been functionally characterized previously. In our study, for the first time, not only their expression in the hypothalamus of rats but also differences in the hypothalamic level of transcription between the hypertensive and normotensive rats were demonstrated and confirmed by qPCR on a large number of RNA samples. Because these lncRNAs belong to the group of four lncRNAs whose transcription correlated with the level of transcription of differentially expressed protein-coding genes, we were able to suggest their possible involvement in the formation of the behavior/neurological phenotype and hypertensive state of ISIAH rats.

  • The idea behind the study is good but the data is from 3 vs 3 rats, which makes it hard to be generalizable. The authors could even look at public resources to increase the number or try specific lncRNA enrichment techniques.

Answer: Since the ISIAH rat strain is the original one and has not been sequenced by other groups, there is simply nothing to combine our data with. The authors understand that the appropriate statistical power is starting from n=6. However, because RNA-sec analysis is expensive, researchers often use less biological replicates in the experiments. Examples are shown below. And we think that these works successfully contribute to the treasury of scientific knowledge.

According to the “Standards, Guidelines and Best Practices for RNA-Seq” from the ENCODE Consortium (http://genome.ucsc.edu/ENCODE/protocols/dataStandards/ENCODE_RNAseq_Standards_V1.0.pdf), experiments should be performed with two or more biological replicates.

So, the RNA-Seq studies published by other groups are often performed with n=2 or n=3:

Zhao S, Fung-Leung WP, Bittner A, Ngo K, Liu X. Comparison of RNA-Seq and microarray in transcriptome profiling of activated T cells. PLoS One. 2014 Jan 16;9(1):e78644. doi: 10.1371/journal.pone.0078644. eCollection 2014. “There were a total of six time points, with two biological replicates per time point.”

Zhongdi Cai, Kaiyue Zhao, Li Zeng, Mimin Liu, Ting Sun, Zhuorong Li,* and Rui Liu*

The Relationship between the Aberrant Long Non-Coding RNA-Mediated Competitive Endogenous RNA Network and Alzheimer’s Disease Pathogenesis. Int J Mol Sci. 2022 Aug; 23(15): 8497.  Published online 2022 Jul 31. doi: 10.3390/ijms23158497. PMCID: PMC9369371

PMID: 35955632 “the RNA-seq analysis was performed by using the tissue samples from three 5×FAD mice and three WT mice (two females and one male per group).

      Only three lncRNAs were statistically significant at p<0.05, yet the authors chose top 6 and two other based on correlation analysis for qPCR. It is not clear how the candidates were chosen? If fold-change was used as criteria, some of them show no fold-change difference between the groups, which makes them bad candidates for validation.

Answer: Yes, indeed only three lncRNAs were statistically significant at p<0.05. We were surprised that there were so few of them, so we analyzed from the beginning of the list until we found differences using real-time PCR. Correlation analysis was performed for all expressed lncRNAs. The discovery of a large number of correlations with differentially transcribed protein-coding genes was sufficient reason for us to test the differential transcription level of RGD1562890 and Tnxa-ps1 by real-time PCR, especially since their fold-change was comparable to that of those lncRNAs, whose expression differences have already been confirmed by us by real-time PCR.

  • The qPCR measurements did not match the RNA-seq results for Tnxa-ps1, then why was it further chosen for correlation-based analysis?

Answer:  From our point of view, the fact that qPCR measurements did not match the RNA-seq results for Tnxa-ps1 is not a critical problem, since the differences obtained in RNA-seq were not significant. If RNA-seq showed an alternative ratio with high statistical probability, this would be a serious problem. In our work, we should trust the results obtained in qPCR, since this analysis was performed on a large number of samples.

Also, it is not clear what happened to the other lncRNAs candidates? How do the authors account for randomness in the correlation analysis?

Answer:  Correlation analysis was performed for all expressed lncRNAs. However, functional annotation was performed only for those lncRNAs for which a correlation was found with a large number of differentially expressed genes. This approach allows 1) to make assumptions about the involvement of lncRNA in certain biological processes and 2) reduces the likelihood of errors in formulating these hypotheses.

  • The network analysis performed by the authors for 4 of the lncRNA co-expressed genes shows highly non-connected genes. The authors highlighted some networks and functions in the figures. Are these the top functions from String or chosen at random by the Authors?

Answer: The terms determined by String and highlighted in the figures were randomly selected by the authors as the most appropriate to characterize features that may be related to the hypertensive phenotype. However, in the revised version of the manuscript we performed functional annotation in the Cytoscape_v3.9.1 and it showed the highlighted terms as the top ones (see figures 4-7).

The enrichment analysis should be performed against the identified genes as background as against all genes (default).

Answer: We made the enrichment analysis against the list of genes identified as expressed in our RNA-seq data (we used this list as a background).

Minor:

  • Reduce the title to “Identification of Long Noncoding RNAs Associated with Hyper-tension and the Behavior/Neurological Phenotype of Hypertensive ISIAH Rats”

Answer: The title was reduced according to your recommendation.

  • It’s not clear from the abstract or text that the study is a validation of earlier study and not done again.

Answer: A clarification was made to the text of the abstract

  • Please mention version numbers of genome annotations used.

Answer: the version number of genome annotations used is given in text (please, see line 106)

  • String figures can be re-done using only connected nodes (and use of tools like Cytoscape)

Answer: Done. Only connected nodes are shown in the Figures 4-7. String networks were sent to the Cytoscape_v3.9.1 using StringApp where the functional annotation for these String networks was performed. The top terms are shown in Figures 4-7.

The authors are grateful to the reviewer for constructive comments, which allowed us to substantially strengthen the manuscript.

Reviewer 2 Report

The manuscript "Identification of Long Noncoding RNAs Coexpressed in the Hypothalamus with mRNAs of Genes Associated with Hypertension and the Behavior/Neurological Phenotype of Hypertensive ISIAH Rats" identified lncRNAs associated with the hypertensive state and behavioral features characteristic of ISIAH rats using RNA sequencing. The manuscript is engaging, and most importantly, the authors' findings are an initial step into understanding the transcriptional regulation of hypothalamic processes contributing to specific hypertensive and neurological phenotypes. Here are some comments to improve the manuscript:

In lines 195-197, the authors mention: In total, 12 lncRNAs correlating with one or more lncRNAs may take part in the regulation of the expression of 176 (91.7%) DEGs, i.e., almost all the identified DEGs (Supplementary Table 3). Did they mean lncRNAs correlating with one or more DEGs? Please verify.

In line 214 and line 359, the authors address the main discrepancy of their study (qPCR results for two lncRNAs, Gas5 and Tnxa-ps1, whose expressions differ from those initially obtained through RNA-seq analysis). They hint that both results did not match due to individual differences in the expression of these lncRNAs in the rat hypothalamus. However, is this the only possible explanation for the discrepancies? The manuscript would benefit from a more elaborate discussion on this point.

Line 322: Figure 8a -> 8A

Lines 406-408 lack references

Lines 419-420: Define Ang II and AT1

Lines 448 and 553: in both instances, the word "confirm" overstates the actual findings. Maybe use "suggest"?

Finally, the manuscript lacks consistency regarding the abbreviations (Abbreviate when the full name first appears). For example, the full names of the lncRNAs identified are described in the discussion section, not when they first appeared. Also, several genes described in the discussion are presented as abbreviations first and then, in parenthesis, in full. 

Author Response

Comments and Suggestions for Authors

The manuscript "Identification of Long Noncoding RNAs Coexpressed in the Hypothalamus with mRNAs of Genes Associated with Hypertension and the Behavior/Neurological Phenotype of Hypertensive ISIAH Rats" identified lncRNAs associated with the hypertensive state and behavioral features characteristic of ISIAH rats using RNA sequencing. The manuscript is engaging, and most importantly, the authors' findings are an initial step into understanding the transcriptional regulation of hypothalamic processes contributing to specific hypertensive and neurological phenotypes. Here are some comments to improve the manuscript:

In lines 195-197, the authors mention: In total, 12 lncRNAs correlating with one or more lncRNAs may take part in the regulation of the expression of 176 (91.7%) DEGs, i.e., almost all the identified DEGs (Supplementary Table 3). Did they mean lncRNAs correlating with one or more DEGs? Please verify.

Answer: thank you very much for this comment. We have corrected this error (please, see line 204).

In line 214 and line 359, the authors address the main discrepancy of their study (qPCR results for two lncRNAs, Gas5 and Tnxa-ps1, whose expressions differ from those initially obtained through RNA-seq analysis). They hint that both results did not match due to individual differences in the expression of these lncRNAs in the rat hypothalamus. However, is this the only possible explanation for the discrepancies? The manuscript would benefit from a more elaborate discussion on this point.

Answer: Since the authors see no other reasons that could explain the discrepancies obtained, we ask the reviewer to forgive us for not being able to expand the discussion on this topic.

Line 322: Figure 8a -> 8A

Answer: Thank you very much. We have made correction.

Lines 406-408 lack references

Answer: the reference was inserted

Lines 419-420: Define Ang II and AT1

Answer: Done. Please, see lines 438-439 in the revised version of the manuscript.

Lines 448 and 553: in both instances, the word "confirm" overstates the actual findings. Maybe use "suggest"?

Answer: Line 448 (line 465 in the revised version of the manuscript) - the word confirm has been replaced with the word suggest

 Line 553 (line 570 in the revised version of the manuscript): - the word confirm has been replaced with Our data are in a good agreement with this point of view“

Finally, the manuscript lacks consistency regarding the abbreviations (Abbreviate when the full name first appears). For example, the full names of the lncRNAs identified are described in the discussion section, not when they first appeared. Also, several genes described in the discussion are presented as abbreviations first and then, in parenthesis, in full. 

Answer: the lncRNAs and reference genes were abbreviated when the full name first appears (lines 137-143 in the revised version of the manuscript)

The accepted version of the manuscript will be sent to the language editing company.

The authors thank the referee for a careful reading of the manuscript and constructive comments, which allowed us to improve the text.
